# Coupling Coordination between Fintech and Digital Villages: Mechanism, Spatiotemporal Evolution and Driving Factors—An Empirical Study Based on China

**Chengkai Zhang, Yanjun Zhang \*, Yu Li and Shan Li**

Beijing Academy of Science and Technology, Beijing 100089, China; zckzck3317@126.com (C.Z.);
liyu_2016@126.com (Y.L.); lishandepp@163.com (S.L.)
**\*** Correspondence: ssapple@126.com

**Abstract:** Based on Chinese provincial data from 2013 to 2020, this research constructed a fintech index and a digital index and analyzed the temporal and spatial coupling coordination status and driving factors of the two using a coupled coordination model. The results of the study were as follows. (1) In general, the comprehensive index of fintech and digital villages increased year by year in the time sequence and fell into fintech-dominated coupling. The divergence in space was significant, showing an overall decreasing trend from the eastern coastal areas to the central areas and western areas. (2) In terms of sequential characteristics, the coupling coordination between fintech and digital villages shifted into the stage of primary coordination, which was phased and rising and continued to grow during the examination period. (3) In terms of spatial characteristics, the degree of coupling coordination between fintech and digital villages was different and agglomerative, with a trend of "strong in the east, mediocre in the middle and poor in the west"; seven provinces and cities entered the intermediate coordination stage. (4) In terms of drivers, the levels of economic development, regional industrial structure, regional population density, and digital infrastructure had a positive influence on coupling coordination. (5) Conditional convergence existed in eastern, central and western China from the convergence test; also, the speed was faster than absolute convergence.

**Keywords:** financial technology; digital villages; coupling coordination degree; driving factors; coordinated development



## 1. Introduction

With the gradual maturation of technologies such as artificial intelligence, metaverse and blockchain, the integration and promotion of digital technologies into various fields is gradually becoming a focus of global attention [1,2]. In China, this focus is occurring in the area of financial and rural digital integration, which is urgently needed for the country's modernization to drive long-term regional growth. As China's GDP per capita rose from USD 1042 in 2001 to USD 12,741 in 2022 and the Engel coefficient fell from 40.5% in 2001 to 29.8% in 2021, the rapid economic development was accompanied by a gradual widening of the urban–rural divide. The countryside has lagged far behind urban development. In addition to the massive backlog of per capita distribution issues, a range of macro issues, such as infrastructure development and rural industrial development, are also important manifestations of the economic growth plaguing rural areas. Therefore, how to make use of the digital dividend brought about by the digital integration of finance and the countryside to promote further development of the countryside has become a difficult issue of concern for scholars and the Chinese government.

Equitable distribution, balanced growth and improved infrastructure are important components of the United Nations Sustainable Development Goals. The digital economy is the main economic form after the agricultural and industrial economies. It has dramatically changed the basic shape of industry and agriculture and is bound to have a huge impact

on rural development. Under the framework of the policy of building a digital China, the Chinese government has proposed a strategic plan for fintech and the digital countryside that aims to promote the deep integration of digital technology with economic, political, cultural, social and ecological civilization in order to achieve Chinese modernization and reach the ultimate goal of common prosperity [3]. Therefore, it can be seen that fintech and the digital countryside have a strong policy fit: on one hand, fintech brings new financial support for the development of the digital countryside, and on the other hand, there is increasing demand for fintech, and the construction of the digital countryside provides a new scenario for fintech development. In the new era of digital economic development, promoting the deep integration of fintech and the digital countryside is an essential instrument to achieve the policy goals of digital China and a key step to bridge the urban–rural divide. Therefore, there is an urgent need for China to explore the specific effects and coordination mechanisms for the integration of fintech and digital rural development. In this context, this study has strong practical significance and policy guidance value.

With the rapid development of fintech and the digital countryside, the interplay between the two has become more complex. This paper will explore the spatial relationship between the integration of the two and provide a comprehensive understanding of the layers and inherent mechanisms of the coordinated development of fintech and digital villages. However, the literature on the integration of fintech and the digital countryside is not sufficient. Most of the existing studies explore the micro-levels within their respective fields, and few articles have explored the interrelationship between fintech and digital villages. Therefore, this paper aims to conduct an in-depth study from the following perspectives. First, the composite index of fintech and digital villages has shown an increasing trend year by year, with obvious spatial and temporal divergence. Second, the coupling and coordination degree of fintech and digital villages shows a phase and rise in time characteristics. Third, the coupling and coordination degree of fintech and digital villages has differences and agglomeration in space. Fourth, the coupling and coordination degree of fintech and digital villages is the result of multiple factors together. Fifth, the coupling and coordination between fintech and digital villages is driven by a combination of factors, and there is spatial heterogeneity.

The marginal contributions of this paper are: first, this paper proposes for the first time the coordination mechanism between fintech and the digital countryside, constructs a comprehensive index evaluation system for fintech and digital villages, and continuously enriches the relevant theories. Second, it uses the coupling coordination degree model and panel regression model to empirically analyze the coupling coordination degree of fintech and digital villages as well as the driving factors affecting the coupling coordination between them, to deeply analyze the mutual promotion relationship between fintech and digital villages and provide a theoretical basis and policy suggestions for their integration and development.

## 2. Literature Review and Mechanism of Coupling Action between Fintech and Digital Villages

### 2.1. Literature Review

In recent years, based on the great achievements of fintech and digital villages, the study of the relationship between the two has been the focus of scholars. Digital villages provide an important opportunity for the development of fintech, providing a digital facility base for the development of fintech, generating diverse financial needs, and reducing financial costs [4]. To further explore the relationship between the two, scholars take a more macro perspective such as "fintech and rural revitalization". Fintech has a natural advantage in promoting rural revitalization. According to Hayek's "supply priority" theory, the primary task of rural revitalization development is to play a leading role in financial supply. Digital finance, as a carrier of inclusive finance, can provide services in demand to customers with almost zero marginal cost immediately, which suggests that this approach is

free from the time and geographical restrictions of traditional financial institutions [5]. The development of rural revitalization also requires a high level of fintech (2021, Jigang Li). The development between digital inclusive finance and rural revitalization is characterized by a "single threshold"; when the development level of digital inclusive finance is lower than a specific threshold, it can positively facilitate rural revitalization. When the development level is higher than this threshold, the promotion effect can be significantly enhanced [6].

Scholars have also conducted extensive exploration in the subfields of fintech and digital villages. Fintech [7–9] (2017, Ma) refers to a range of digital technologies that widely influence financial payments, financing, lending, investment, financial services, and money operations [10]. With a strong innovation and revolutionary effect, fintech (2018, Pi Tianlei) has exploited emerging technologies to enhance the efficiency of financial services, revolutionize financial markets, create new financial products, and expand the demand for financial services, giving rise to new business models, as well as shaping the ways in which to acquire credit, risk rating, and pricing [11]. There is a deep-rooted mechanism for the development of fintech (2022, Shi Zonghui), which is to promote the "shift of financial resources from virtual to real", allocate more financial resources to medium and high customers and agriculture-related enterprises, and achieve not only the inclusive growth of economic development and benefits for disadvantaged groups but also the harmonious growth of the economy and the environment, as well as more welfare for the people [12]. Based on exploring the connotation and mechanism of finance, scholars have expanded their research to other fields with empirical studies focusing on the construction of the Digital Financial Inclusion Index (2020, Guo Feng) [13]. First, regarding the development of common prosperity [14,15] (2022, Sun Jiguo), fintech can significantly contribute to the revitalization of rural industries and achieve common prosperity by alleviating farm-related financing constraints, promoting the circulation of agricultural land and enhancing agricultural technology innovation [16]. Digital inclusive finance can promote the growth of consumption and achieve common prosperity by building a long-term mechanism to broaden access to credit and promote human capital accumulation [17] (2022, Yan Jingrui). Fintech can contribute to the achievement of common wealth through three paths, namely, "increasing productivity", "sustainable development" and "reducing inequality" through the financial growth effect, financial sustainability effect and financial deepening effect, respectively [18]. Second, with a focus on the exploration of industrial development [19,20], (2022, Li Youshu), fintech significantly promotes the upgrading of regional industrial structure; this is the case even after considering that the upgrading of industrial structure eliminates endogenous problems and creates a stronger path dependence [21] (2022, Li Haiqi). By increasing entrepreneurial opportunities, encouraging scientific and technological innovation, alleviating financing constraints, and narrowing the gap between urban and rural areas, industrial structure optimization and industrial upgrading in China can be effectively promoted [22].

Digital villages [23–25] are a modernized development and transformation process of agriculture and rural areas. Such development is endogenous to the application of networking, informatization and digitization in the economic and social development of agriculture and rural areas, as well as the improvement of farmers' modern information skills (2021, Zeng Yiwu). In terms of content, digital villages can be divided into five dimensions: digital infrastructure, rural data resource development and management, rural digital industrialization, rural industry digitization, and rural governance digitization [26]. (2022, Xie Wenshuai) In terms of mechanisms, digital village construction is profoundly changing rural production and lifestyles by promoting the liberation and development of rural digital productivity with data element empowerment, the change in rural production relations with the form of platform-based economic organization, and the adjustment of rural superstructure with the digital transformation of rural governance [27]. Based on the level of connotation and mechanism [28], scholars have explored the specific content relating to the construction of digital villages. Digital infrastructure is the foundation and primary task of digital village construction (2022, Li Guixin). The government needs to

continuously increase financial investments in the construction of digital infrastructure related to fiber optics and communication to narrow the dual pattern of infrastructure construction between urban and rural areas [29]. In the face of data development and management in the construction of digital villages (2021, Sun Xiang), applied information technology based on a big data platform that can effectively sense, process and analyze systematic application scenarios in the countryside, which can in turn support multiuser customization, the multidimensional analysis of data and multi-industry service expansion, provides new solutions for rural industrial and economic development and improves farmers' quality of life and grassroots governance capacity [30]. Studies have shown that (2022, Wang Xiaona) digital village construction has not only given a strong impetus to rural development but also unblocked the flow of urban and rural elements and products and promoted the development of rural e-commerce; furthermore, the overall quality of industrial development has shown a favorable trend of continuous improvement [31]. On the one hand, momentum can empower industrial prosperity (2022, Li Benqing). The construction of digital villages can significantly promote the level of industrial prosperity, and its impact on each dimension of industrial prosperity varies, with more obvious promotion effects present in the dimension of high-quality agricultural development, the dimension of industrial integration development and the dimension of shared development of farmers [32]. On the other hand, with innovative and effective rural governance, the operation of rural governance in digital space has changed the content and form of rural governance, prominently reflected in the improvement of villagers' consultation and self-governance, the promotion of diversified governance powers, and the construction of villagers' collective identity. The construction of digital villages uses digital information technology to reconstruct traditional rural governance, prompting the empowerment of rural governance subjects, the innovation of governance methods and the rebuilding of governance communities.

In summary, the definition and measurement of fintech or the digital village has been relatively well researched, but there is less literature that directly addresses the relationship between fintech and digital villages. The vast majority of the existing literature focuses on the contribution of fintech to rural revitalization and common wealth within their respective subfields, as well as the construction of digital infrastructure, rural digital industries and rural digital governance in the digital countryside, and lacks the perspective of empirical research to explore the relationship between the two. Therefore, this paper attempts to fill the gaps in four aspects: (1) supplementing and improving the intrinsic mechanism of the coupled and coordinated development of fintech and the digital countryside; (2) measuring the degree of coupled and coordinated development of fintech and the digital countryside; (3) analyzing the temporal and spatial characteristics of the coupled and coordinated relationship; (4) empirically analyzing the influence of various factors on the degree of coupled and coordinated development of the two systems.

### 2.2. Analysis of the Mechanism of Coupling Action between Fintech and Digital Villages

Fintech, as an emerging form of financial development, and digital villages, as a typical representative of rural revitalization strategy, are interactively coupled with the optimization and improvement of internal factors. Based on that, the author believes that fintech can promote the rapid development of digital villages, and the construction of digital villages can complement the development of fintech, two processes that are closely linked and interactively coordinated with each other.

### 2.2.1. Analysis of the Mechanism of the Role of Fintech in Digital Villages

Fintech is a new form of finance featuring digitization and platforms. It breaks the traditional financial model of serving the countryside and promotes the further development of digital villages by expanding financial supply, consolidating infrastructure construction and bridging the digital divide [33]. First, fintech can effectively promote the development of new infrastructure and data integration in the countryside. With the support of artificial

intelligence, 5G, cloud computing and other digital technologies, data information has realized information sharing and utilization, thereby solving the problem of lack of rural information from the source, prompting more capital to flock to the rural basic data field, enriching rural basic data to form a positive cycle, and providing solid support for the new infrastructure construction of digital villages. By using agricultural-related databases, including "farmer household information, transaction records, credit records and production information", fintech can provide inclusive financial services around various digital application scenarios in the field of "agriculture, rural areas and farmers". With the operations of mining big data, synthesizing user profiles and providing personalized and precise services, fintech has promoted the digitization of agricultural production, rural management and farmers' lives and provided fintech assistance for the integration and development of digital rural development strategies. Second, fintech can effectively expand the financial supply of digital agriculture. Based on the "three pain points" of rural finance derived from the phenomenon of systemic negative investment, i.e., "high service cost, information asymmetry, and lack of collateral", traditional financial institutions are limited in their scale of supply to agriculture-related fields and are less willing to conduct business. As a result, it is difficult to cover the capital needs of medium and high customers and agricultural enterprises (2021, Wang Junshan) [34]. By establishing mechanisms for data sharing, precise services, commercial sustainability and scenario extension, fintech has solved the "three major pain points" of rural finance and injected new momentum into rural life and agricultural production. For example, farmers can use financial tools to obtain large amounts of capital to attract talent in agricultural technology, update agricultural digital technology, expand the scale of agricultural production, and enrich the accumulated agricultural production factors, further promoting the development of digital agriculture. Third, fintech can provide financial convenience for rural residents through new financial scenarios. The energy of financial inclusion is released with the help of digital technology, which objectively enhances the ability of rural residents to use modern financial tools such as internet finance and mobile payments. On the one hand, such inclusion bridges the gap of the "digital divide" between urban and rural areas, breaks the spatial–geographical boundaries through digital technology, reduces the information barriers of farmers, enriches the digital usage scenarios of rural areas, and achieves the overall improvement of residents' digital financial literacy (2022, Liu Shaojie) [35]. On the other hand, it realizes the crossing of the "digital divide" between urban and rural areas, and with the power of fintech, it guides agricultural-related financial institutions to provide the same convenient financial services to urban and rural residents through new scenarios of innovative digital financial products.

### 2.2.2. Analysis of the Mechanism of the Role of Digital Villages in Fintech

Along with the commercial maturity and popular application of new-generation digital information technology, the digital economy has become an important part in leading China's social and economic development. As an important support for rural revitalization and the Digital China strategy, the construction of digital villages will greatly promote the development of rural areas and provide strategic development opportunities for fintech (2022, Yi Jun) [36]. First, the construction of digital villages provides the necessary infrastructure environment for fintech. The promotion of services such as financial assistance to farmers, mobile payments and remote loans in rural areas has been a disadvantage for agricultural-related financial institutions, which is attributed to the weak network infrastructure in rural areas. The construction of digital villages can compensate for this shortcoming. On the issue of improving rural network infrastructure, the construction of backbone networks, metropolitan area networks and local area network facilities will greatly enhance the rural digital network environment, accelerate the popular application of rural digital financial products and enhance the ability of rural residents to use digital technology, all of which provide the necessary basic conditions for the development of rural fintech. Second, the construction of digital villages creates a demand for financial diversity.

From the overall perspective of rural development, with the construction of digital villages, the digital level of rural areas is significantly improved, and digital technologies such as artificial intelligence and big data are being widely used in agricultural product production, farm operation, rural education, rural medical care, rural pensions and other fields, thereby boosting the digital upgrade of the countryside and generating a large number of financial needs of enterprises. For rural residents, digital technology allows them to access a wide range of personal financial services, such as financial management, personal loans, agricultural insurance and other financial services, thus giving rise to diverse individual financial needs and providing a wide pool of customers for the development of rural digital inclusive finance. Moreover, the construction of digital villages can reduce the cost of financial services. The construction of digital villages can enhance the use of mobile terminals and electronic computers by rural residents and accelerate the promotion of fintech in rural areas. Such construction can significantly reduce the cost for financial institutions to provide financial transactions and further break the limitations of traditional physical outlets for financial services, thus enhancing the rate of financial service utilization.

### 2.2.3. Analysis of the Mechanisms for the Integration of Fintech and Digital Village Development

In summary, fintech promotes the further development of digital villages by means of government intervention and market intervention in three aspects: namely, expanding financial supply, consolidating infrastructure, and bridging the digital divide. Digital villages complement fintech development in three aspects, namely, building a digital environment, generating financial demand, and reducing financial costs by means of social development and industrial development. The final result is the benign resonance and coordinated development of the coprosperity and mutual promotion of fintech and digital villages. Therefore, this paper proposes hypothesis H1: There is a state of coupled coordination between fintech and digital villages, with temporal and spatial variability. The interactive development mechanism of the coupling of the two is shown in Figure 1.

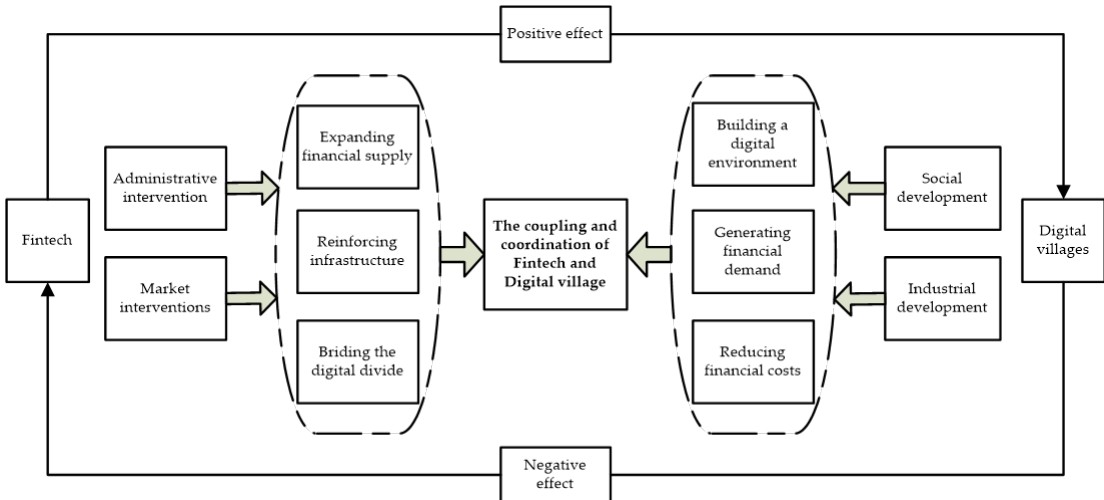

**Figure 1.** Coupling mechanism of fintech and digital village development.

A summary of existing studies found that the degree of coordination between fintech and digital village coupling is driven by multiple factors, among which the level of regional economic development, regional industrial structure, regional population density, and the level of digital infrastructure [37–40] are the most prominent. The higher the economic level of the region, the higher the demand for digital enhancement, and when digitization is enhanced, the reverse stimulates sustained economic growth. Theoretically, the higher the level of economic development, the more significantly the degree of coupling between the two can be improved. Regional industrial restructuring is crucial to the coupling of fintech

and digital villages, while the variable of regional population density is an important driver of the coupling of the two systems from the perspective of population space. The degree of digital infrastructure development in the region as a whole is an important factor influencing the coordination of the coupling of fintech and digital villages, which involves the coverage effect, stability and speed of operation. Therefore, this paper proposes hypothesis H2: The coupling and coordination of fintech and the digital countryside is influenced by the level of regional economic development, regional industrial structure, regional population density, and the level of digital infrastructure. Considering the disparity between the economic and regional rural development in the east, middle and west of China leading to differences in the degree of development of fintech and the digital countryside, it is necessary to conduct an inter-regional variability analysis (Zhang Yanyan, 2022) [41]. There are obvious regional differences in the total economic development and regional rural construction in China; especially, the total economic share has gradually decreased in the east, central and west, but the scale of rural areas is gradually increasing in the east, central and west (Zhou, Haipeng, 2016) [42]. There is also a phenomenon of regional concentration of financial resources, with eastern cities ranking first in terms of excessive concentration of financial resources, while western cities rank at the bottom in terms of the level of financial resource aggregation (Su, Fang, 2016) [43], and financial exclusion is more serious in remote areas of central and western China. Therefore, hypothesis H3: There is regional heterogeneity in the impact of multiple factors on the coupling and coordination of fintech and digital villages.

## 3. Indicator System, Research Methods and Data Sources

### 3.1. The Construction of the Indicator System

To more objectively, comprehensively and systematically analyze the coupling and synergy between China's fintech and digital village construction and comprehensively consider the representativeness of the indicators, the following indicator systems are proposed in this paper. See Table 1 for details.

**Table 1.** Comprehensive indicator system for fintech and digital villages.

| Systems | Tier 1 Indicators | Secondary Indicators | Interpretation of Secondary Indicators (Units, Attributes) | Weighting | Data Sources |
|---|---|---|---|---|---|
| Fintech | Breadth of coverage Depth of use Degree of digitization | - | - | - | Peking University Digital Inclusive Finance Index (2011–2020) |
| Digital villages | Financial input | Investment in agricultural production | Amount of investment in agriculture in the category of rural fixed assets as a proportion of the total output value of agriculture, forestry, animal husbandry and fisheries (%, positive) | 0.033 | China Rural Statistical Yearbook |
| | | Investment in IT applications in agriculture | Investment in fixed assets in rural transport, storage and postal services (RMB billion, positive) | 0.076 | China Rural Statistical Yearbook |
| | Infrastructure | Rural basic communication facilities | Length of fiber optic cable lines (km, positive) | 0.043 | China Statistical Yearbook |
| | | Rural internet communication facilities | Number of rural broadband access subscribers (million, positive) | 0.069 | China Statistical Yearbook |
| | | Rural mobile communication facilities | Mobile phone penetration rate per 100 population (units) | 0.023 | China Statistical Yearbook |
| | | Rural radio and television communication facilities | Number of rural cable broadcast TV subscribers as a proportion of total households (%, positive) | 0.039 | China Statistical Yearbook |

**Table 1.** *Cont.*

| Systems | Tier 1 Indicators | Secondary Indicators | Interpretation of Secondary Indicators (Units, Attributes) | Weighting | Data Sources |
|---|---|---|---|---|---|
| Digital villages | Agricultural production | Application of science and technology in agricultural production | Number of national modern agricultural demonstration zones and industrial parks, number of national demonstration parks for integrated development of rural industries and agricultural science and technology parks and number of national key leading enterprises in agricultural industrialization (pcs, positive) | 0.034 | Ministry of Agriculture and Rural Affairs. National Development and Reform Commission |
| | | Scale of agricultural production | Number of farms as a proportion of primary sector output (%, positive) | 0.071 | China Rural Statistical Yearbook |
| | | Electricity consumption for rural production | Rural electricity consumption (billion kWh, positive) | 0.108 | China Statistical Yearbook |
| | | Agricultural machinery applications | Total power of agricultural machinery (million kW, positive) | 0.052 | China Statistical Yearbook |
| | | Agrometeorological applications | Number of operational agrometeorological observation sites (nos., positive) | 0.021 | China Statistical Yearbook |
| | | Digital agriculture talent | Number of agricultural technicians (persons, positive) | 0.035 | National Statistical Office; Provincial Statistical Yearbooks |
| | Life services | Rural e-commerce penetration | Percentage of Taobao villages among all administrative villages (%, positive) | 0.263 | The 1% Change—2020 China Taobao Village Research Report; administrative village statistics from the National Bureau of Statistics |
| | | Rural e-commerce transaction value | E-commerce purchases and sales (billion) | 0.100 | China Statistical Yearbook |
| | | Rural logistics coverage | Rural delivery routes (km, positive) | 0.033 | China Statistical Yearbook |

Fintech indicator system. Regarding the construction of fintech indicators, the author referred to the design idea of Guo Feng et al. and used the logitized digital finance index as a representative indicator of fintech. Among them, fintech encompasses the range of coverage, depth of use, and digitization of three mostly financial big data types, which can develop a more objective and comprehensive index evaluation for the current situation of fintech development in China and is now widely used in the relevant literature (2020, Huang Rui) [44].

Digital villages index system. First, with "digital villages" as the keyword, the CNKI database was used to conduct frequency statistics of 622 documents related to digital villages from 2000 to 2022, and the indicators with higher frequency were selected. Relevant indicators in the Digital Agriculture and Rural Development Plan (2019–2025), China Digital Villages Development Report (2020), Digital Villages Standard System Construction Guide and other official plans and reports were also selected. Finally, a number of experts were interviewed and consulted, and a digital village indicator system was constructed based on their opinions. A total of 15 specific indicators were developed in four dimensions, namely, financial input, infrastructure, agricultural production and living services, with indices ranging from 0 to 1.

*3.2. Data Sources*

The data related to the fintech indicators came from the Peking University Digital Inclusive Finance Index (2011–2020). The data related to digital village indicators and the factors influencing the degree of coupling and coordination came from the China Statistical Yearbook, China Rural Statistical Yearbook, and statistical yearbooks of various provinces, including the data in the 1% Change—2020 China Taobao Village Research

Report. Among the collected data, for the problem of missing data for some specific indicators of individual provinces or regions in the statistical yearbook, this research acquired data from the National Bureau of Statistics, the National Development and Reform Commission, the Ministry of Agriculture and Rural Affairs, websites of provincial and municipal statistical bureaus, and government work reports and used the interpolation method to impute the missing data that could not be found. In addition, Tibet, Hong Kong, Macao, and Taiwan were excluded due to serious levels of missing data.

### 3.3. Research Methodology

The purpose of this paper was to measure the development level of fintech and digital villages in China by constructing an indicator system of fintech and digital villages, analyze the spatial difference effect of coupling coordination between regions in China by using a coupling coordination model, and estimate the driving factors affecting coupling coordination by using panel data regression. The specific steps and methods used to do so are described as follows.

### 3.3.1. Standardization of Index Data

As there are different dimensional units among the indicators, the raw data were standardized for uniformity and comparability across the indicators.

The specific methods are as follows:

$$\text{Positive indicators}: \lambda_{ij} = \frac{x_{ij} - minx_j}{maxx_j - minx_j} \tag{1}$$

$$\text{Negative indicators}: \lambda_{ij} = \frac{maxx_j - x_{ij}}{maxx_j - minx_j} \tag{2}$$

where $i$ represents the province, and $j$ represents the indicator. The value of $\lambda_{ij}$ obtained through Formulas (1) and (2) forms the index matrix $(\lambda_{ij})_{mn}$, where $m$ denotes the number of sample identities, and $n$ denotes the number of evaluation indicators.

### 3.3.2. Entropy Evaluation Method

This research measured the level of development of fintech and digital villages by standardizing the raw data and then conducting weighted summation. To evaluate the utility of each indicator more objectively, the entropy evaluation method was used to determine the weights of each indicator. The specific operation steps are described as follows.

In the first step, the weight $p_{ij}$ of each indicator $\lambda_{ij}$ in the fintech system and the digital village system was calculated:

$$p_{ij} = \frac{\lambda_{ij}}{\sum_{i=1}^{m} \lambda_{ij}} \tag{3}$$

In the second step, the information entropy $e_j$ and information effect value $d_j$ were calculated under the $j$th indicator:

$$e_j = -\frac{1}{ln(m)}\sum_{i=1}^{m} p_{ij}lnp_{ij} \tag{4}$$

$$d_j = 1 - e_j \tag{5}$$

In Formula (5), $1 - e_j$ indicates the coefficient of variation of the selected indicators in the $j$th indicator, and its value reflects the magnitude of the impact of the data evaluation of the indicator.

In the third step, the comprehensive weight $\xi_j$ of each indicator was calculated:

$$\xi_j = \frac{d_j}{\sum_{j=1}^{n} d_j} \tag{6}$$

In the fourth step, a comprehensive evaluation index $I_{ij}$ was calculated by summing the combined weight $\xi_j$ of each indicator with the corresponding standardized $\lambda_{ij}$:

$$I_{ij} = \sum_{i=0}^{n} \xi_j \lambda_{ij} \tag{7}$$

With reference to the index system, it is possible to calculate the level of development of digital inclusive finance $K_1$ and the level of development of digital villages $K_2$ in turn.

### 3.3.3. Coupling Coordination Model

The first model is the coupling degree model. Coupling is used to reflect the relationship of mutual influence and interaction between two or more systems. To study the mutual influence relationship between fintech and digital villages, this research established the coupling degree model of fintech and digital villages based on the above evaluation index system (2021, Li Na) [45] with the following formula:

$$C = \sqrt{\frac{K_1 K_2}{(K_1 + K_2)^2}} \tag{8}$$

In Equation (8), $C$ represents the level of coupling between fintech and digital village development and takes values in the range of [0, 1] [46].

The second model is the coordination degree model. To deeply study the coordination relationship between fintech and digital villages and reflect the high or low level of coupling between the systems, a coordination degree model was introduced; the specific model is as follows:

$$T = \alpha K_1 + \beta K_2 \tag{9}$$

In Equation (9), $T$ represents the degree of coordination between fintech and digital villages, where $\alpha$ and $\beta$ are coefficients to be determined. This research considered fintech and digital villages equally important. Therefore, $\alpha + \beta = 0.5$.

The third model is the coupling and coordination degree model. Since the single coupling degree model and coordination degree model cannot fully portray the coordinated relationship between fintech and digital villages, this research further constructed a coupling coordination degree model to objectively reflect the degree of coupling development and the coordination effect of the two systems, as follows:

$$D = \sqrt{CT} \tag{10}$$

In Equation (10), $D$ indicates the degree of coupled and coordinated development of fintech and digital villages, and the value range is [0, 1]. To more intuitively reflect the level of coupling and coordination between fintech and digital villages, based on the actual development degree of the two systems and the research of scholar Tan Yanzhi, the mean segmentation method was used as the basis to divide $D$ into eight levels [45], as shown in Table 2.

**Table 2.** Fintech and digital village coupling coordination development level.

| Coupling Coordination Degree | Coupling Coordination Level | Coupling Coordination Degree | Coupling Coordination Level |
|---|---|---|---|
| (0, 0.1) | Extremely imbalanced | (0.4, 0.5) | Intermediate coordination |
| [0.1, 0.2) | Severely imbalanced | [0.5, 0.6) | Benign coordination |
| [0.2, 0.3) | On the verge of imbalance | [0.6, 0.8) | High coordination |
| [0.3, 0.4) | Primary coordination | [0.8, 1.0) | High-quality coordination |

3.3.4. Panel Data Regression Model

In order to examine the influence of the many influencing factors on the coupling of the two systems, this paper constructed the model:

$$D_{it} = \beta_0 + \beta_1 EDL_{it} + \beta_2 RIS_{it} + \beta_3 RPD_{it} + \beta_4 DIL_{it} + \xi FE + \phi FE + \varepsilon_{it} \qquad (11)$$

where the subscripts $i$ and $t$ are individual and time, respectively; $\beta_0$ is a constant term; $\beta_1 - \beta_4$ are the regression coefficients of each variable; $\xi FE$ and $\phi FE$ represent fixed time and individual effect, respectively; and $\varepsilon_{it}$ is the error term.

In order to further analyze the influence of various factors on the degree of coordination between fintech and digital village coupling, this paper used the degree of coordination between fintech and digital village coupling (*D*) calculated above as the explanatory variable, and selected the level of economic development, regional industrial structure, regional population density and the level of digital infrastructure as explanatory variables, respectively.

Among them: (1) economic development level (*EDL*). Referring to Liu Yaobin's (2007) [37] paper, GDP per capita is used to express the level of economic development. Both fintech and digital villages cannot be developed without the continuous growth of national and local economies, and a high level of economic form helps the development of both, and it is expected that this variable will positively influence the degree of coordination of fintech and digital village coupling; (2) regional industrial structure (*RIS*) [38]. Referring to Lu Fengping (2017), the ratio of the value added of the secondary industry to the value added of the primary industry is used to express the regional industrial structure, optimizing the regional industrial restructuring and strongly supporting the construction of the digital countryside to achieve high-quality rural development, and it is expected that this variable will negatively influence the degree of coordination of the coupling of fintech and digital countryside; (3) regional population density (*RPD*). Referring to Wang Yi (2023), the year-end resident population to land area is used to indicate regional population density [39]. A high-density population will bring more digital demand and enrich the development of fintech and digital villages, and it is expected that this variable will positively influence the degree of coordination of fintech–digital village coupling; (4) digital infrastructure level (*DIL*). Referring to She Maoyan (2021) [40], the length of fiber-optic cable lines is used to indicate the level of digital infrastructure. The level of this indicator directly reflects the overall digital intensity of the region, and both fintech and the digital countryside are products of digital development to a certain extent, and this variable is expected to positively influence the degree of coordination of fintech–digital countryside coupling.

## 4. Empirical Study

### 4.1. Analysis of the Comprehensive Index of Fintech and Digital Villages

4.1.1. Time-Order Characteristics of the Comprehensive Index

A comprehensive index of fintech and digital villages of 30 provinces in China from 2013 to 2020 was acquired after systematic measurement, as shown in Figure 2. On the whole, fintech and digital villages in all provinces of China are developing rapidly with a progressive rise, but the overall development trend of fintech is better than the development of digital villages. From the perspective of fintech development, the level of development in 2013 was similar to that of digital villages, but fintech development maintained an

average growth rate of 11.94% and continued to grow, peaking at 0.342 in 2020. From the perspective of the development of digital villages, the average annual growth rate from 2013 to 2020 was 7.49%. Despite its continuous annual growth, the development of digital villages has been much slower than that of fintech. The main reasons are as follows. The first reason is the rapid gathering of development factors. With the advent of the internet era, the influx of talent, technology and capital-based elements into the digital field has realized a great explosion of fintech, allowing the rapid popularization and application of finance in China. The second reason is that the digital foundation is too poor. The digital infrastructure in the vast rural areas of China is still in its infancy, and the digital development of the countryside is relatively poor and requires long-term development. The third reason is the role of national policies. China is strongly supporting the development of fintech and digital villages. Various policies have been introduced from the Central Committee of the Communist Party of China and the State Council to various local governments to address the weak links between fintech and digital villages, providing a favorable policy environment for their development.

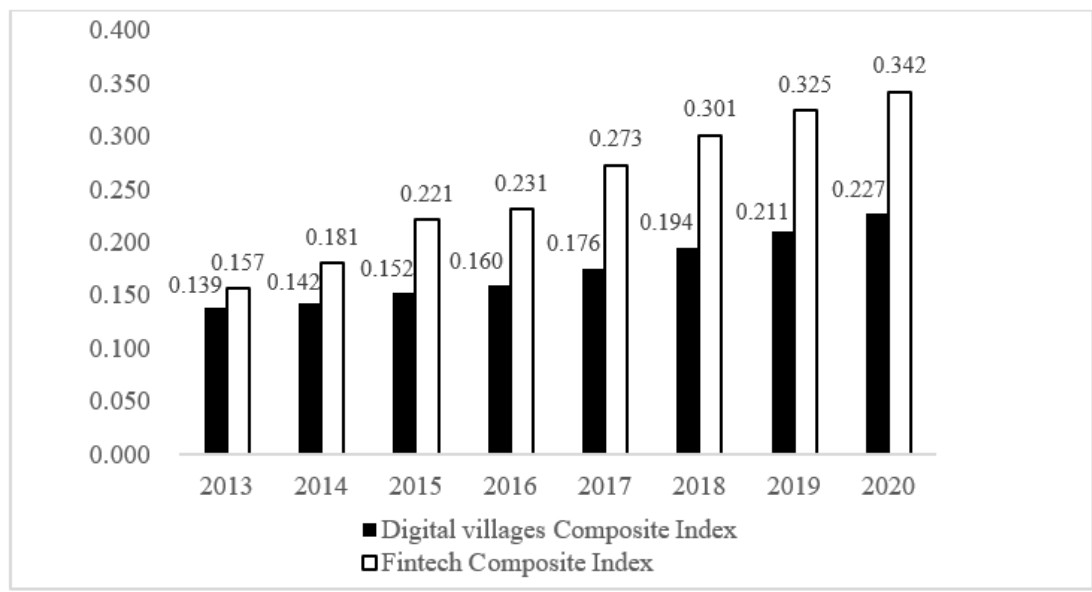

**Figure 2.** Time-series characteristics of the fintech and digital villages composite index.

4.1.2. Spatial Characteristics of the Comprehensive Index

Fintech has been characterized by significant spatial aggregation, with a central development area of "one belt and three cores" located mainly in eastern provinces and cities and a secondary development area of "one pole and multiple points" supported in inland regions, as shown in Table 3. The formation of the eastern development economic zone has given rise to the Circum-Bohai Sea region, with "Beijing-Tianjin" as the fintech core; the Yangtze River Delta region, with "Zhejiang-Shanghai-Jiangsu" as the fintech core; and the southeast region, with "Guangdong-Fujian" as the fintech core. The provinces where the three geographic centers are located had a comprehensive fintech index value between 0.380 and 0.432 in 2020, with a mean value of 0.394. Among them, Beijing and Shanghai had the highest fintech index values, objectively reflecting the investment of the two cities in factor supply, industrial development and policy support and making them strategic locations to drive the development of fintech nationwide. The central provinces, with Hubei Province as the fintech core, have become the new growth pole of the inland region, with a comprehensive index value of 0.359 for fintech, thereby driving the formation of a multipoint-coordinated development pattern in the central and western regions and other inland provinces. The provinces with lower levels of fintech development are mainly located on the northern border of China, ranging from Heilongjiang Province in the east to

Xinjiang Uygur Autonomous Region in the west, with an overall fintech index value of less than 0.350.

**Table 3.** Fintech and digital village development level.

| Region | 2013 | | 2015 | | 2017 | | 2020 | |
|---|---|---|---|---|---|---|---|---|
| | Fintech | Digital Villages | Fintech | Digital Villages | Fintech | Digital Villages | Fintech | Digital Villages |
| National | 0.157 | 0.139 | 0.221 | 0.152 | 0.273 | 0.176 | 0.342 | 0.227 |
| Beijing | 0.216 | 0.114 | 0.276 | 0.126 | 0.330 | 0.155 | 0.418 | 0.239 |
| Tianjin | 0.175 | 0.062 | 0.238 | 0.066 | 0.284 | 0.080 | 0.362 | 0.112 |
| Hebei | 0.145 | 0.190 | 0.200 | 0.219 | 0.258 | 0.227 | 0.323 | 0.300 |
| Shanxi | 0.144 | 0.129 | 0.206 | 0.141 | 0.260 | 0.142 | 0.326 | 0.130 |
| Inner Mongolia | 0.147 | 0.112 | 0.215 | 0.117 | 0.259 | 0.129 | 0.309 | 0.148 |
| Liaoning | 0.160 | 0.153 | 0.226 | 0.155 | 0.267 | 0.168 | 0.326 | 0.164 |
| Jilin | 0.138 | 0.132 | 0.208 | 0.128 | 0.255 | 0.140 | 0.308 | 0.145 |
| Heilongjiang | 0.141 | 0.131 | 0.210 | 0.141 | 0.257 | 0.147 | 0.306 | 0.177 |
| Shanghai | 0.222 | 0.155 | 0.278 | 0.180 | 0.337 | 0.195 | 0.432 | 0.274 |
| Jiangsu | 0.181 | 0.294 | 0.244 | 0.352 | 0.298 | 0.415 | 0.382 | 0.557 |
| Zhejiang | 0.206 | 0.210 | 0.265 | 0.266 | 0.318 | 0.354 | 0.407 | 0.574 |
| Anhui | 0.151 | 0.134 | 0.211 | 0.147 | 0.272 | 0.162 | 0.350 | 0.238 |
| Fujian | 0.183 | 0.140 | 0.245 | 0.163 | 0.299 | 0.206 | 0.380 | 0.291 |
| Jiangxi | 0.146 | 0.114 | 0.208 | 0.124 | 0.267 | 0.160 | 0.341 | 0.173 |
| Shandong | 0.159 | 0.282 | 0.221 | 0.319 | 0.272 | 0.369 | 0.348 | 0.430 |
| Henan | 0.142 | 0.196 | 0.205 | 0.216 | 0.267 | 0.218 | 0.341 | 0.257 |
| Hubei | 0.165 | 0.158 | 0.227 | 0.144 | 0.285 | 0.178 | 0.359 | 0.199 |
| Hunan | 0.148 | 0.137 | 0.206 | 0.155 | 0.261 | 0.181 | 0.332 | 0.213 |
| Guangdong | 0.185 | 0.254 | 0.241 | 0.291 | 0.296 | 0.401 | 0.380 | 0.582 |
| Guangxi | 0.142 | 0.106 | 0.207 | 0.122 | 0.262 | 0.129 | 0.325 | 0.190 |
| Hainan | 0.158 | 0.046 | 0.230 | 0.051 | 0.276 | 0.051 | 0.344 | 0.058 |
| Chongqing | 0.160 | 0.058 | 0.222 | 0.071 | 0.276 | 0.083 | 0.345 | 0.118 |
| Sichuan | 0.153 | 0.166 | 0.216 | 0.178 | 0.268 | 0.220 | 0.335 | 0.332 |
| Guizhou | 0.121 | 0.081 | 0.193 | 0.085 | 0.252 | 0.103 | 0.308 | 0.146 |
| Yunnan | 0.138 | 0.118 | 0.204 | 0.116 | 0.256 | 0.131 | 0.319 | 0.179 |
| Shaanxi | 0.148 | 0.110 | 0.216 | 0.109 | 0.267 | 0.132 | 0.342 | 0.156 |
| Gansu | 0.128 | 0.080 | 0.200 | 0.092 | 0.244 | 0.099 | 0.306 | 0.112 |
| Qinghai | 0.118 | 0.057 | 0.195 | 0.064 | 0.240 | 0.065 | 0.298 | 0.070 |
| Ningxia | 0.137 | 0.061 | 0.215 | 0.056 | 0.256 | 0.062 | 0.310 | 0.081 |
| Xinjiang | 0.143 | 0.183 | 0.206 | 0.174 | 0.249 | 0.172 | 0.308 | 0.170 |

The digital villages index in China has formed a "digital development belt" along the eastern coast and shows a gradient decline along the east and west. In the "digital development belt" of the eastern coast, Guangdong Province, Zhejiang Province, Jiangsu Province and Shandong Province show the most remarkable performance, with their comprehensive index values reaching 0.582, 0.574, 0.557 and 0.430, respectively, making them the leading regions in the east. The low values in the comprehensive index of digital villages mainly include the central provinces, represented by Henan Province, with comprehensive index values ranging from 0.257 to 0.130. In contrast to their level of fintech development, some western provinces have become concentrated areas of low values on the comprehensive index of digital village development; the values of other western provinces are not higher than 0.190, except for Sichuan Province, which is better developed.

Based on the geographical pattern in Figure 3, the two comprehensive indices can be seen to form a decreasing trend from the coastal areas to the inland areas, with obvious spatial links in the distribution characteristics and a close relationship with provincial economic and social development. Moreover, the two comprehensive indices of some provinces have shown performance that extends beyond the region. For example, Hubei Province, which has better fintech development, reached a value of 0.359, and Sichuan Province, which has better development of digital villages, reached a value of 0.332, denoting them as the leaders that drive the development of the surrounding regions.

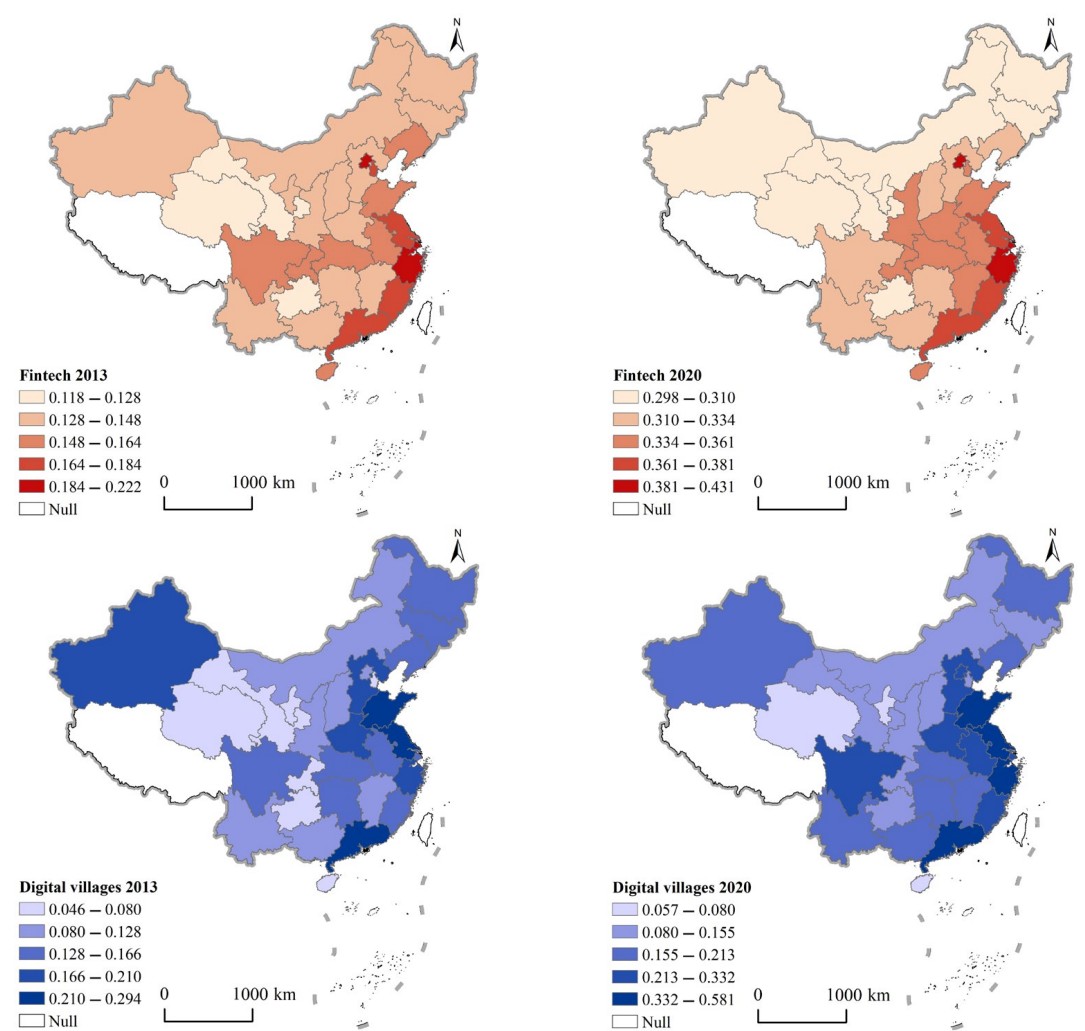

**Figure 3.** Spatial distribution of the fintech and digital villages index (2013 and 2020).

### 4.2. *Time-Order Characteristics Analysis of Coupling Coordination between Fintech and Digital Villages*

4.2.1. Time-Order Characteristics of the Coupling Coordination Degree

Under the condition of determining the comprehensive development level of fintech and digital villages, Equation (10) was applied to further measure the coupling coordination degree of the two systems, and the results are shown in Figure 4.

The mean value of fintech–digital village coupling coordination is shown to have grown continuously during the examination period, i.e., from 0.266 in 2013 to 0.362 in 2020; furthermore, it is shown to have entered the primary coordination stage, indicating that the mutual influence between the two systems of fintech and digital villages is still not profound enough. From the perspective of the amplitude of change, the mean value of coupling coordination has increased each year, with an average annual growth rate of 4.5%. This indicates that the interaction and mutual influence between systems are continuously improving, but there is still huge room for growth on the whole. The reasons are as follows. First, the country has vigorously boosted the overall development of the digital economy sector, prompting the overall support of factors, policies and the environment for areas related to the digital economy. Second, there is still an aspect of uneven spatial distribution in the development of fintech, and the development of digital villages is still subject to the shackles of the current separate urban-rural structure, leading to a limited increase in the mean value of coupling coordination.

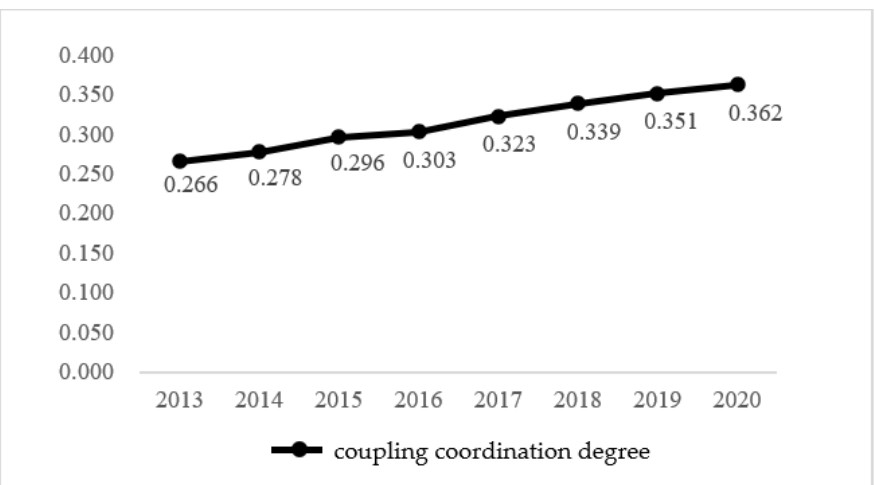

**Figure 4.** Time-series characteristics of the coupling coordination degree between fintech and digital villages.

There are phased characteristics of the mean value of coupling coordination, which grew rapidly from 2013 to 2015 but suddenly slowed down between 2015 and 2016, with only slow growth of approximately 6.6% until the beginning of 2017. The construction of Digital China has always been an important strategic direction for the development of China. The policy on fintech and digital villages issued by the Central Committee of the CPC and the State Council has scientifically guided the standardized development of the financial industry and the construction of the countryside and pointed to the direction for the coordinated development of fintech and digital villages. Additionally, digital development has received attention from local governments, and relevant policies have been introduced to support and guide it. For example, around 2016, Beijing, Shanghai, Zhejiang Province, Jiangsu Province, and Guangdong Province all introduced development strategies, plans and action plans related to fintech and digital villages, which greatly contributed to the continuous growth of coupling coordination between fintech and digital villages. Thus, the hypothesis is confirmed by the existence of a coupled and coordinated state of fintech and digital villages, with temporal and spatial differentiation.

4.2.2. Spatial Characteristics of Coupling Coordination Degree

The spatial characteristics of the coupling coordination degree of fintech and digital villages from 2013 to 2020 were analyzed, as shown in Table 4. In 2013, the spatial distribution of the coupling coordination degree of fintech and digital villages was extremely uneven, and the coupling coordination degree of each province and city ranged from 0.203 to 0.340, with a mean value of 0.266. There were 25 provinces and cities at the stage of endangered dissonance, mainly concentrated in the central, western and partly eastern regions, including Beijing, Tianjin, Chongqing and other regions with better traditional digital development. There were five provinces and cities in the primary coordination stage, including Jiangsu Province, Guangdong Province, Shandong Province and other regions, among which Jiangsu Province had the highest coupling coordination degree of 0.340. The mean values of coupling coordination in the east, central and west were 0.290, 0.266 and 0.240, respectively, forming a decreasing trend from east to west. Therefore, it can be seen that the level of coupling coordination between fintech and digital villages was very low in 2013, and the overall level was in the initial stage.

**Table 4.** Fintech and digital village development levels.

| Region | Province | 2013 | Ranking | 2020 | Ranking |
|---|---|---|---|---|---|
| Eastern | Zhejiang | 0.322 | 4 | 0.492 | 1 |
| | Guangdong | 0.329 | 2 | 0.485 | 2 |
| | Jiangsu | 0.340 | 1 | 0.480 | 3 |
| | Shandong | 0.325 | 3 | 0.440 | 4 |
| | Shanghai | 0.305 | 5 | 0.415 | 5 |
| | Fujian | 0.283 | 10 | 0.408 | 7 |
| | Beijing | 0.280 | 12 | 0.398 | 8 |
| | Hebei | 0.288 | 7 | 0.394 | 9 |
| | Liaoning | 0.279 | 13 | 0.340 | 18 |
| | Tianjin | 0.228 | 24 | 0.317 | 26 |
| | Hainan | 0.207 | 29 | 0.266 | 30 |
| Central | Henan | 0.289 | 6 | 0.385 | 10 |
| | Anhui | 0.267 | 14 | 0.380 | 11 |
| | Hubei | 0.284 | 9 | 0.366 | 12 |
| | Hunan | 0.266 | 15 | 0.365 | 13 |
| | Jiangxi | 0.254 | 19 | 0.348 | 15 |
| | Heilongjiang | 0.261 | 16 | 0.341 | 17 |
| | Inner Mongolia | 0.253 | 20 | 0.327 | 21 |
| | Jilin | 0.260 | 18 | 0.325 | 23 |
| | Shanxi | 0.261 | 17 | 0.321 | 24 |
| Western | Sichuan | 0.282 | 11 | 0.408 | 6 |
| | Guangxi | 0.247 | 23 | 0.353 | 14 |
| | Yunnan | 0.253 | 22 | 0.345 | 16 |
| | Shaanxi | 0.253 | 21 | 0.340 | 19 |
| | Xinjiang | 0.284 | 8 | 0.338 | 20 |
| | Guizhou | 0.222 | 26 | 0.326 | 22 |
| | Chongqing | 0.220 | 27 | 0.318 | 25 |
| | Gansu | 0.225 | 25 | 0.304 | 27 |
| | Ningxia | 0.213 | 28 | 0.281 | 28 |
| | Qinghai | 0.203 | 30 | 0.269 | 29 |

Compared with its level in 2013, the coupling coordination degree between fintech and digital villages in 2020 (Table 4) realized a large increase in each province and city, with a significant difference between provinces and cities. The maximum and minimum values of the coupling coordination degree in 2020 for each province and city were 0.492 and 0.266, respectively, and the mean value was 0.362. Three provinces and cities represented by the Ningxia Hui Autonomous Region and Qinghai Province were still on the verge of disorder, with a coupling coordination degrees of less than 0.3. In general, the agglomeration of various elements has resulted in the unbalanced objective status of the two systems of fintech and digital villages, and the coupling coordination degree of both is still in the unbalanced development stage.

It shows that the overall spatial characteristics of the coupling coordination between fintech and digital villages are influenced by the differences in economy, society, policy and local conditions. According to the Figure 5, although the coupling coordination was not high at the beginning of the examination period, some eastern areas had entered the primary coordination stage. The high-value aggregation area that emerged at the end of the examination period was mainly located in the eastern coastal area, and the spatial distribution showed a trend of decline from east to west. Therefore, hypothesis H1: The development of coupled and coordinated fintech and digital villages grows over time and shows spatial variability, is confirmed.

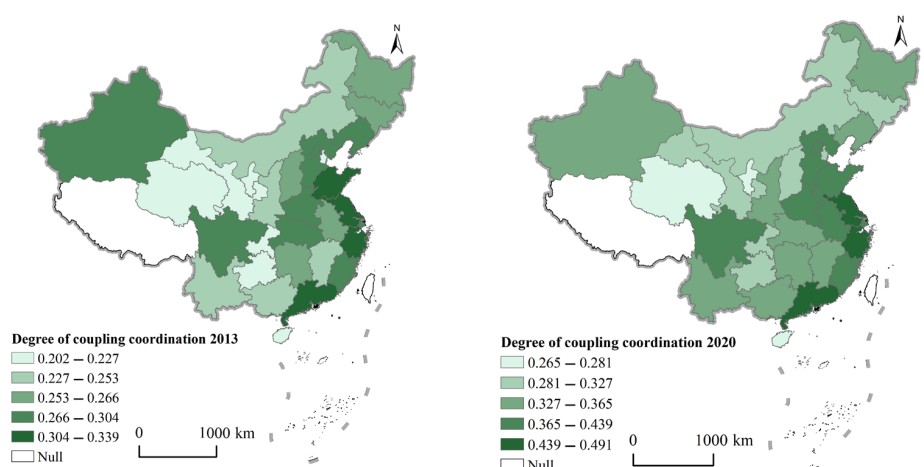

**Figure 5.** Spatial distribution of fintech and digital village coupling coordination (2013 and 2020).

### 4.3. Study of Driving Factors

#### 4.3.1. Descriptive Statistical Analysis

Table 5 reports the results of descriptive statistics for the main research variables in this paper. It can be seen that among all 240 variables, the maximum and minimum values of the explanatory variables fintech and digital villages coupling coordination (D) were 0.49 and 0.20 respectively, with a standard deviation of 0.06, indicating that there were large differences in the level of coupling coordination in different regions. The explanatory variables economic development level (EDL), regional industrial structure (RIS), regional population density (RID) and digital infrastructure level (DIL) had different variability, where the maximum value of economic development level (EDL) was 0.49, the minimum value was 0.02, and the standard deviation was 0.03, with a certain degree of fluctuation. Regional industrial structure (RIS) had a standard deviation of 0.16, with a maximum value of 0.99 and a minimum value of 0.01, indicating that there were large differences in the industrial structure of the provinces. Regional population density (RPD) had a standard deviation of 0.72, with a maximum value of 3.95 and a minimum value of 0.01, indicating an uneven distribution of population density across the country, with some regions having a higher population density. Digital infrastructure level (DIL), with a standard deviation of 0.83, a maximum value of 3.99 and a minimum value of 0.07, reflected the differences in the level of digital infrastructure across provinces. The performance of the explanatory variables was generally consistent with the results of previous studies and has been detailed in the explanation of variables above and will not be repeated here.

**Table 5.** Descriptive statistical analysis.

| Variables | Sample Size | Mean | Median | Standard Deviation | Minimum Value | Maximum Value |
|---|---|---|---|---|---|---|
| D | 240.00 | 0.31 | 0.31 | 0.06 | 0.20 | 0.49 |
| EDL | 240.00 | 0.06 | 0.05 | 0.03 | 0.02 | 0.16 |
| RIS | 240.00 | 0.09 | 0.05 | 0.16 | 0.01 | 0.99 |
| RPD | 240.00 | 0.48 | 0.29 | 0.72 | 0.01 | 3.95 |
| DIL | 240.00 | 1.13 | 0.91 | 0.83 | 0.07 | 3.99 |

#### 4.3.2. Correlation Study

Table 6 shows the results of the variable correlation tests. Preliminarily, it shows that the degree of coupled coordination between fintech and digital villages has a strong positive correlation with economic development level (EDL), a positive correlation with regional industrial structure (RIS) and regional population density (RPD), and the strongest positive correlation with digital infrastructure level (DIL), but the sign of the regression coefficient and the degree of correlation may change due to fixed effects, and the preliminary analysis

still needs to be regressed for further validation. Further variance inflation factor (VIF) test results show that the value of the VIF between the variables was 8.85 and less than 10, indicating that the multicollinearity between the variables in the empirical model of this paper was not strong and could be carried out in the next step of regression analysis.

**Table 6.** Correlation study.

| Variables | D | EDL | RIS | RPD | DIL |
|-----------|-----|-----|-----|-----|-----|
| D | 1.0000 | | | | |
| EDL | 0.5712 *** | 1.0000 | | | |
| RIS | 0.2065 *** | 0.7636 *** | 1.0000 | | |
| RPD | 0.2722 *** | 0.7134 *** | 0.9479 *** | 1.0000 | |
| DIL | 0.8522 *** | 0.2273 *** | −0.1587 ** | −0.0473 *** | 1.0000 |

"***", "**" indicate significance at the 1%, 5% levels of significance, respectively.

### 4.3.3. Regression Analysis

The OLS model was used to carry out regression estimation of the impact of four major factors, namely the level of economic development, regional industrial structure, regional population density and the level of digital infrastructure, on the degree of coordination of the coupling between the two systems of fintech and digital villages. The *p*-value of the Hausman test was less than 0.01, and the original hypothesis was rejected; thus a two-way fixed effects model was used. The regression results are shown in Table 7. The hypothesis was verified through the hierarchical regression method, and the results of each variable added for estimation from column (1) to column (4) showed significance at the 1% confidence level, indicating the influence of the four factors of economic development level, regional industrial structure, regional population density and digital infrastructure level on the degree of coordination of the coupling between the two systems of fintech and the digital countryside. Hypothesis H2 was verified.

**Table 7.** Regression analysis.

| | (1) | (2) | (3) | (4) |
|---|---|---|---|---|
| | D | D | D | D |
| edl | 1.051 *** | 1.853 *** | 1.387 *** | 1.230 *** |
| | (0.356) | (0.404) | (0.250) | (0.229) |
| ris | | −0.174 *** | −0.179 *** | −0.105 *** |
| | | (0.0525) | (0.0293) | (0.0299) |
| rpd | | | 0.653 *** | 0.463 *** |
| | | | (0.165) | (0.159) |
| dil | | | | 0.0178 *** |
| | | | | (0.00277) |
| Time effect | Control | Control | Control | Control |
| Individual effects | Control | Control | Control | Control |
| Constant | 0.220 *** | 0.199 *** | −0.0872 | −0.00713 |
| | (0.0168) | (0.0148) | (0.0757) | (0.0703) |
| Observations | 240 | 240 | 240 | 240 |
| R-squared | 0.926 | 0.937 | 0.958 | 0.970 |
| Number of pro | 30 | 30 | 30 | 30 |

Note: (1) Standard deviation values are in brackets; (2) "***" indicate significance at the 1% levels of significance.

### 4.3.4. Endogeneity Test

In addition to the endogeneity problem caused by the omission of common shock variables, the findings of this paper will be affected by endogeneity caused by two-way causality, as the degree of coupled coordination between fintech and digital village is not only the result of the underlying research, but may also act as an independent variable to drive the development of the four major factors. To solve this problem, this paper used the four major influencing factors of economic development level, regional industrial structure, regional population density and digital infrastructure level with one period lag as the instrumental variables, respectively, and then used 2SLS to retest the research construction, and the corresponding regression results are shown in Table 8. According to the full-sample regression results, the estimated coefficients of the four major influencing factors were still positive at the 1% significance level, indicating that they promoted the degree of coordination between fintech and digital village coupling, and hypothesis H2 is further corroborated.

**Table 8.** Endogeneity test.

| Variables | (1) | (2) | (3) | (4) | (5) | (6) | (7) | (8) |
|---|---|---|---|---|---|---|---|---|
| | edl | D | ris | D | rpd | D | dil | D |
| l.edl | 0.8689 *** (0.038) | | | | | | | |
| l.ris | | | 0.8178 *** (0.129) | | | | | |
| l.rpd | | | | | 0.8314 *** (0.0914) | | | |
| l.dil | | | | | | | 0.8544 *** (0.049) | |
| edl | | 1.725 *** (0.186) | 0.9523 *** (0.414) | 1.564 *** (0.152) | −0.218 * (0.1305) | 1.496 *** (0.161) | −1.704 (2.048) | 1.492 *** (0.152) |
| ris | 0.0167 *** (0.005) | −0.142 *** (0.0249) | | −0.130 *** (0.0290) | 0.0296 (0.041) | −0.124 *** (0.0256) | −0.342 (0.303) | −0.106 *** (0.0248) |
| rpd | −0.0181 (0.016) | 0.524 *** (0.0824) | −0.115 (0.120) | 0.540 *** (0.0867) | | 0.614 *** (0.107) | 0.031 (0.968) | 0.497 *** (0.0859) |
| dil | 0.0005 (0.0004) | 0.0167 *** (0.00219) | −0.0034 (0.003) | 0.0168 *** (0.00218) | 0.0046 ** (0.002) | 0.0161 *** (0.00223) | | 0.0209 *** (0.00252) |
| Time effect | Control | Control | Control | Control | Control | Control | Control | Control |
| Individual effects | Control | Control | Control | Control | Control | Control | Control | Control |
| Constant | 0.0810 0.0617 | −1.801 *** (0.313) | 0.5028 (0.473) | −1.852 *** (0.330) | 0.6707 * (0.3439) | −2.136 *** (0.409) | 0.591 (3.690) | −1.698 *** (0.329) |
| Observations | 210 | 210 | 210 | 210 | 210 | 210 | 210 | 210 |
| R-squared | 0.998 | 0.989 | 0.995 | 0.990 | 1.000 | 0.989 | 0.979 | 0.989 |

"***", "**" and "*" indicate significance at the 1%, 5% and 10% levels of significance.

### 4.3.5. Heterogeneity Test

In order to explore the heterogeneity of each influencing factor, namely the level of economic development, regional industrial structure, regional population density, and the level of digital infrastructure, on the degree of coordination of the coupling between fintech and the digital countryside, the sample data were divided into regions for analysis. Table 9 shows the results of the regional heterogeneity of the impact of each influencing factor on the degree of coordination of the coupling between fintech and the digital countryside by dividing the sample data into eastern, central, and western regions. Overall, the impact of each factor on the degree of coordination of the coupling between fintech and digital villages was significant in the eastern region, while partial insignificance existed in the central and western regions. In terms of impact coefficients, the impact coefficients of all variables, except regional population density, were significantly higher in the eastern region than in the central and western regions. The possible reasons for this were that the

eastern region has a more developed economy, a higher standard of living for rural residents, and strong financial demand from enterprises and residents, while the progressive development of financial technology has improved financial accessibility and enhanced the efficiency of production and life for enterprises and resident groups, making them eager for the convenience brought by the construction of the digital village, fueling the promotion of each element to the coupling of financial technology and the digital village in the eastern region. The economic development disadvantages of the western region are obvious, and the industrial structure and population density are not dominant, and the rural residents have a more urgent need for financial digital infrastructure, which inhibits the factors in promoting the coordinated development of the coupling of fintech and the digital countryside. Although the central region has a large population and a certain digital foundation, the overall economic level is not high, and the digital infrastructure needs to be updated and coupled with the high population density, and the government needs to provide a large amount of financial support to improve the rural landscape. The improvement of each factor cannot be directly reflected in the fintech–digital rural coupling index. Therefore, the geographical variation effect of each factor in H3 on the coordination of fintech–digital village coupling was verified, indicating the strong robustness of the core findings of this paper.

**Table 9.** Heterogeneity test.

| Variables | (1)<br>Eastern Region<br>D | (2)<br>Central Region<br>D | (3)<br>Western Region<br>D |
|---|---|---|---|
| edl | 1.408 *** | 0.0578 | 0.431 |
| | (0.285) | (0.335) | (1.139) |
| ris | −0.111 *** | −0.0344 | −0.256 |
| | (0.0313) | (0.205) | (0.538) |
| rpd | 0.355 ** | 0.826 *** | 0.846 |
| | (0.145) | (0.121) | (0.833) |
| dil | 0.0245 *** | 0.0141 * | 0.0180 *** |
| | (0.00587) | (0.00625) | (0.00457) |
| Time effect | Control | Control | Control |
| Individual effects | Control | Control | Control |
| Constant | −0.108 | 0.0408 | 0.113 |
| | (0.120) | (0.0323) | (0.101) |
| Observations | 96 | 72 | 72 |
| R-squared | 0.979 | 0.976 | 0.975 |
| Number of pro | 12 | 9 | 9 |

"***", "**" and "*" indicate significance at the 1%, 5% and 10% levels of significance.

*4.4. Convergence Analysis*

4.4.1. Convergence Test

It can be considered that there is $\sigma$ convergence if the standard deviation of the coupling coordination degree between fintech and the digital countryside in different regions decreases with time. The $\sigma$ coefficient calculation formula is as follows:

$$\sigma_t = \sqrt{\frac{1}{n}\sum_{i=1}^{n}\left(lnD_{it} - \frac{1}{n}lnD_{it}\right)^2} \tag{12}$$

where represents the pair value of coupling coordination degree of a province in a year. If it indicates that the difference of high-quality letter sending levels in different regions decreases with time, then there is convergence. As shown in Figure 6, the coupling and coordinated development coefficient of fintech and the digital countryside decreased somewhat from 2013 to 2020, but it was still in a trend of continuous growth. Therefore, there was no convergence in the coupling and coordination of fintech and the digital

countryside. However, in terms of time and region, the coefficients of eastern and central regions would slow down in 2020, and there was a possibility of subsequent convergence.

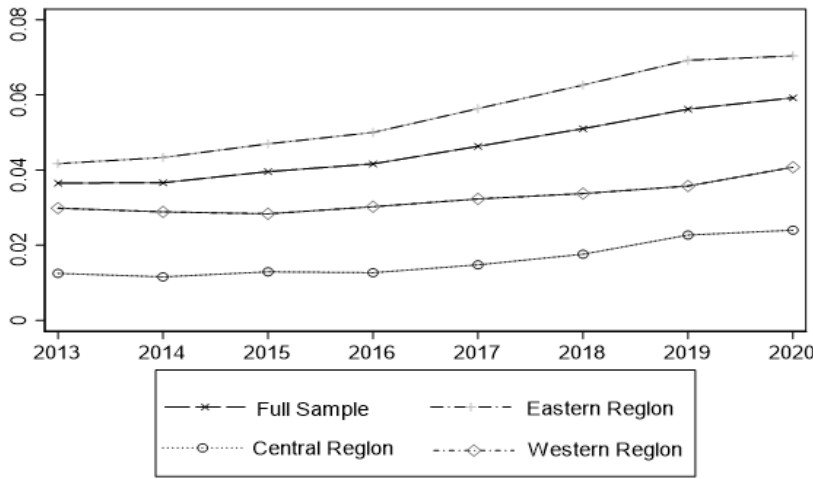

**Figure 6.** Result of convergence test.

### 4.4.2. $\beta$ Convergence Test

$\beta$ convergence examines whether the coupling and coordinated development of fintech and the digital countryside in different development stages show the same growth trend, which can be divided into $\beta$ absolute convergence and conditional $\beta$ convergence. $\beta$ absolute convergence means that the coupling coordination degree of fintech and the digital countryside in backward areas converges to the coupling coordination degree of developed areas and finally reaches the same growth rate. Conditional $\beta$ convergence refers to the condition of convergence depending on other factors in this paper, with reference to Barro and Sala–I–Martin (1992) [47], respectively, to investigate China's financial technology and digital coupling. The coordinated development of rural absolute convergence and conditional convergence model is as follows:

$$\frac{inD_{i,t-1} - lnD_{i,t-1}}{T} = \alpha + \beta lnD_{i,t-1} + \varepsilon_{i,t} \tag{13}$$

$$\frac{lnD_{i,t-1} - lnD_{i,t-1}}{T} = \alpha + \beta lnD_{i,t-1} + X_{i,t} + \varepsilon_{i,t} \tag{14}$$

where $\ln D_{i,t-1}$ is the logarithmic value of coupling coordination degree of province i in year $t-1$; T is the length of time. $X_{i,t}$ is a series of control variables affecting the coupling and coordinated development of fintech and the digital countryside; $\varepsilon_{it}$ is a random disturbance term. If $\beta < 0$ is significant, it indicates that there is $\beta$ convergence in the development of the coupling coordination degree between fintech and the digital countryside in China, and the convergence rate is $v = \frac{-\ln(1+\beta)}{T}$.

As shown in Table 10, there is no absolute convergence in the coupling and coordinated development of fintech and the digital countryside in China on the whole, but absolute convergence exists in the central region. This indicates that the growth rate gap between fintech and digital countryside coupling and coordinated development in China is widening with a slow central region. To further expand the research (variable sources are shown above), this paper selected economic development level, regional industrial structure, regional population density, and digital infrastructure level as control variables to investigate the conditional convergence of the coupling and coordinated development of fintech and the digital countryside. As shown in Table 11, conditional convergence exists at the national level and in the eastern, central and western regions; also, the conditional convergence rate is faster than the absolute convergence rate.

**Table 10.** Results for β absolute convergence.

| Convergence Coefficient | Full Sample | | Eastern Region | | Central Region | | Western Region | |
|---|---|---|---|---|---|---|---|---|
| | OLS | FE | OLS | FE | OLS | FE | OLS | FE |
| β | 0.0380 *** | −0.0175 | 0.0538 *** | 0.0101 | −0.0333 | −0.0675 ** | −0.0045 | −0.0320 |
| | (3.924) | (−1.155) | (3.756) | (0.410) | (−1.356) | (−2.597) | (−0.211) | (−1.208) |
| Constant | 0.0021 | 0.0191 *** | −0.0018 | 0.0127 | 0.0223 *** | 0.0327 *** | 0.0135 ** | 0.0212 *** |
| | (0.682) | (4.086) | (−0.370) | (1.549) | (2.953) | (4.107) | (2.240) | (2.864) |
| Observations | 210 | 210 | 84 | 84 | 63 | 63 | 63 | 63 |
| F | 15.40 *** | 1.33 | 14.11 *** | 0.17 | 1.84 | 6.75 ** | 0.04 | 1.46 |

t/z-statistics in parentheses *** $p < 0.01$, ** $p < 0.05$.

**Table 11.** Results for β conditional convergence.

| Convergence Coefficient | Full Sample | | Eastern Region | | Central Region | | Western Region | |
|---|---|---|---|---|---|---|---|---|
| | OLS | FE | OLS | FE | OLS | FE | OLS | FE |
| β | −0.0938 *** | −0.4238 *** | −0.0911 ** | −0.5425 *** | −0.1964 *** | −0.3251 *** | −0.1225 ** | −0.2972 *** |
| | (−3.909) | (−7.985) | (−1.991) | (−5.601) | (−4.637) | (−3.842) | (−2.274) | (−2.709) |
| edl | 0.0819 ** | 0.8876 *** | 0.0552 | 1.1328 *** | 0.0292 | 0.4246 | −0.0320 | 0.3974 |
| | (2.241) | (6.029) | (0.894) | (4.889) | (0.370) | (1.421) | (−0.253) | (0.944) |
| ris | 0.0149 | −0.0462 ** | 0.0382 * | −0.0652 ** | −0.0695 * | 0.0984 | 0.0109 | 0.3419 |
| | (1.168) | (−2.020) | (1.821) | (−2.062) | (−1.790) | (0.660) | (0.112) | (1.369) |
| rpd | −0.0014 | 0.2145 *** | −0.0067 * | 0.1790 ** | 0.0114 ** | 0.2740 | 0.0059 | 0.5640 |
| | (−0.587) | (3.494) | (−1.767) | (2.087) | (2.205) | (1.266) | (0.670) | (1.418) |
| dil | 0.0078 *** | 0.0106 *** | 0.0090 *** | 0.0175 *** | 0.0084 *** | 0.0095 ** | 0.0084 *** | 0.0100 *** |
| | (5.523) | (4.923) | (3.177) | (4.013) | (3.389) | (2.438) | (3.788) | (2.966) |
| Constant | 0.0276 *** | −0.0198 | 0.0290 *** | −0.0657 | 0.0605 *** | 0.0027 | 0.0383 *** | −0.0253 |
| | (5.327) | (−0.740) | (2.820) | (−0.932) | (5.651) | (0.047) | (3.229) | (−0.419) |
| Observations | 210 | 210 | 84 | 84 | 63 | 63 | 63 | 63 |
| F | 53.73 *** | 14.39 *** | 22.90 *** | 7.38 *** | 28.66 *** | 5.17 *** | 28.66 *** | 4.29 *** |

t/z-statistics in parentheses. *** $p < 0.01$, ** $p < 0.05$, * $p < 0.1$.

## 5. Conclusions and Recommendations

In this paper, the coupling relationship between the interaction of fintech and digital villages was systematically investigated. Based on that investigation, a comprehensive evaluation index of fintech and digital villages was constructed, and the degree of coupling coordination between fintech and digital villages and the driving factors affecting the coupling coordination were analyzed by using the coupling coordination degree model and a multinomial panel regression model.

### 5.1. Conclusions

First, the comprehensive indices of fintech and digital villages showed an increasing trend year by year, and the temporal and spatial divergence was obvious. In terms of temporal characteristics, the comprehensive indices of both factors continued to increase during the examination period, the comprehensive index of fintech made a greater contribution than the comprehensive index of digital villages did, and the coupling coordination between the systems was dominated by fintech. From the spatial pattern, fintech formed a central development area of "one belt and three cores" located mainly in eastern provinces and cities and a secondary development area of "one pole and multiple points" supported in inland regions. Digital villages formed a "digital development belt" along the eastern coast, with a gradient decline from east to west.

Second, the degree of coupling coordination between fintech and digital villages has been characterized by phases and ascendancy over time. The phased characteristics were reflected in the fact that after 2016, the growth of coupling coordination between systems was significantly higher than that of the previous period. The overall coupling coordination between systems was shown to have a continuous upward trend, which developed from the stage of being on the verge of disorder to the stage of primary coordination. However, there was still a huge growth disparity with regard to quality coordination.

Third, the coupling coordination degree between fintech and digital villages was found to be different and agglomerated in space. Due to the differences in the level of economic

development, population, natural endowment and digital infrastructure construction between regions, the spatial variability between fintech and digital villages was found to be significant, and the spatial distribution of the coupling coordination degree was found to be uneven, with the trend of "strong in the east, mediocre in the middle and poor in the west". The 30 examined provinces and cities across the country showed a large span of coupling coordination, covering the range of verging on coordination to intermediate coordination.

Fourth, the coupling coordination degree of fintech and digital villages was the result of multiple factors working together. From the total sample of 30 provinces, the driving effect of the four factors of economic development level, regional industrial structure, regional population density and digital infrastructure level was the most obvious. Fifth, conditional convergence existed at the national level and in the eastern, central and western regions; also, the conditional convergence rate was faster than the absolute convergence rate.

*5.2. Policy Recommendations*

First, it is necessary to strengthen the construction of digital infrastructure and enhance the development of financial technology and digital villages. On one hand, learn from the US experience in building a series of agriculture-related databases, such as the PESTBANK database, BIOSISPREVIEW database, AGRIS database and AGRICOLA database, and build China's agricultural information service network, supported by satellite networks, the internet and remote sensing networks, to provide digital services for villages in different regions. On the other hand, promote the construction and promotion of financial technology facilities in rural areas, improve the digital financial accessibility of rural residents in less-developed areas, promote the improvement of financial literacy of rural residents, and enhance financial technology capabilities from the intrinsic attributes of residents.

Second, it is important to create a growth pole in the central and western regions and promote the further synergistic development of fintech and digital villages. As the coupling coordination between fintech and digital villages in China is "strong in the east, mediocre in the middle and poor in the west", the primary task is to achieve nationwide coupling and synergetic development to make up for the shortcomings in the middle and west. Learning from the experiences of Japan and Germany, it is necessary to develop unmanned farms and digital total solutions for rural services with agricultural robots as the core, in order to cope with the problem of rural exodus and technological backwardness in the central and western regions and create new growth points.

Third, the mechanism of factor flow should be improved, and internal and external environments that are adapted to the development of fintech and digital villages should be built. Drawing on the role of the market mechanism guided by the concept of "green and sustainable development" in developed countries, it is recommended that a public–private partnership be formed between the government and private companies. Therefore, it is necessary to further promote the market-oriented reform of factors, improve the institutional mechanism of market-oriented allocation and prices through market competition, regulate the relationship between supply and demand, and optimize the allocation of resources. Ultimately, data, population, technology, capital and other important elements that promote the development of fintech and digital villages should be integrated into the practice of digital construction in a scientific and reasonable manner.

## 6. Research Limitations and Discussion

This study focused on the financial integration of fintech and digital villages in the Chinese context, and drew some valuable conclusions. Although this paper largely achieved its original intended research objectives, there were still limitations and the conclusions drawn may be limited by the following three factors, which also point the way for our future research.

First, from the perspective of the research sample, the empirical research in this research focused on China and lacked empirical research on other countries. The conclusions drawn were also geographically limited, and although comparisons were made in terms

of the level of development of the digital village/fintech, there was still a lack of overall research. Future research should be expanded to uphold a more open research perspective, focusing on Asia and conducting in-depth comparative studies on regions such as Europe, the Americas, Africa and Oceania.

Second, from the perspective of research structure, this paper analyzed the spatial and temporal characteristics of the coupling between fintech and digital villages and empirically explored the influencing factors that affect the coupling between the two, but did not conduct an in-depth study on how the influencing factors affect the channels of the coupling between the two, nor did it explore whether there are non-linear characteristics. In the future, there is a need to explore in more depth the level of economic development, regional industrial structure, regional population density, financial activity of residents and digital infrastructure from these two perspectives using the mediation model and the threshold model.

Third, although this study focused on the coupled and coordinated development of fintech and digital villages, the variables affecting the digital development of villages are not limited to fintech, but may also include other elements. In the future, we will try to conduct multi-dimensional research from the perspectives of residents' education, government finance and enterprise transformation, in the hope of further expanding the research on digital villages.

Fourth, the mechanism of the role of fintech and the digital countryside we attempted to construct in this paper is still in the exploratory stage and is not perfect. In the future, a scientific and systematic mechanism should be further constructed to maximize the rationality and credibility of the research results.

**Author Contributions:** Conceptualization, C.Z. and Y.L.; methodology, C.Z.; software, C.Z.; validation, C.Z. and Y.L.; formal analysis, C.Z. and Y.L.; investigation, C.Z. and Y.L.; resources, C.Z., Y.L., S.L. and Y.Z.; data curation, C.Z.; writing—original draft preparation, C.Z. and Y.L.; writing—review and editing, Y.Z., Y.L. and S.L.; visualization, C.Z. and Y.L.; supervision, Y.Z.; project administration, Y.Z.; funding acquisition, C.Z. and Y.Z. All authors have read and agreed to the published version of the manuscript.

**Funding:** This research was funded by the Beijing Academy of Science and Technology municipal financial project "Research on the coordinated development of Beijing-Tianjin-Hebei digital economy" (1420238669KF001-02); "Research on Application Scenario-Driven Market Allocation of Data Elements in Beijing" (1420238669KF001-03).

**Institutional Review Board Statement:** Not applicable.

**Informed Consent Statement:** Not applicable.

**Data Availability Statement:** The data used to support the findings of this study are available from the corresponding author upon request.

**Conflicts of Interest:** The authors declare no conflict of interest.

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
