# Peer review of "Coupling Coordination between Fintech and Digital Villages: Mechanism, Spatiotemporal Evolution and Driving Factors—An Empirical Study Based on China"

_sustainability, doi:10.3390/su15108265_

Round 1

Reviewer 1 Report

This is a high-quality research paper. I recommend acceptance.

Author Response

感谢您在百忙之中阅读本文并对作者的研究表示认可。祝您周末愉快!

Reviewer 2 Report

First of all, I would like to thank you for the possibility of reviewing this interesting paper that I have read with great interest.

The paper may have a clear interest associated to researchers from different scientific disciplines and, therefore, could have a notable repercussion in specialized scientific literature.

Why is this study necessary? should make clear arguments to explain what the originality and value of the proposed model is. This should be stated in the final paragraphs of introduction and conclusion sections.

Literature overview

I would like to suggest the following references:

Fülöp, M. T., Topor, D. I., Ionescu, C. A., Căpușneanu, S., Breaz, T. O., & Stanescu, S. G. (2022). Fintech accounting and Industry 4.0: future-proofing or threats to the accounting profession?. Journal of Business Economics and Management23(5), 997-1015.

Akram, U et al. (2021). Impact of digitalization on customers’ well-being in the pandemic period: Challenges and opportunities for the retail industry. International Journal of Environmental Research and Public Health, 18(14), 7533.

Method is well

Results and discussion are well

Conclusions: pleas add theoretical, managerial, and practical implications, limitation and further research. Some parts are included but must be extended.

, the authors should make an additional effort to solve the problems previously mentioned.

However, I hope that all these comments will serve the author to improve the quality of the paper. Finally, I hope that the comments will be understood positively by the authors of this interesting paper.

Good luck!

Author Response

Thank you for taking the time to read this article and expressing your recognition of the author's research.

Below, we will answer your comments and questions one by one:

Firstly, regarding the issues in the introduction, we have revised the structure and content, incorporating the display background of the coupling and coordination between financial technology and digital rural areas into the introduction. Based on theory and practice, we have proposed the importance and practical significance of the article, as well as its innovation and contribution points.

Secondly, regarding the issues in the literature review, we have thoroughly studied the important literature you provided and added it to our references. Thank you again for providing the important literature.

Thirdly, regarding the issues in the conclusion, based on the comments of the reviewers, the research significance of this article has been added to the conclusion section, and a new section has been listed to discuss the limitations of the research and future directions.

Reviewer 3 Report

I appreciate the documentation made in order to develop this article proposal in favorable terms.

I appreciate the application part of this article proposal through which the authors tested and validated their model, finally concluded and made recommendations.

Considering the above, I recommend the publication of this article proposal.

Author Response

Thank you for reading this article amidst your busy schedule and expressing your recognition of the authors' research. Wishing you a pleasant weekend!

Reviewer 4 Report

The manuscript sustainability-2325838 is devoted to the actual scientific problem, namely study of  peculiarity of fintech and digital villages. The reviewed article is interesting for scholars and theme of the article meets the scope of the journal. Work is performed at sufficient scientific level and has good quality. The manuscript may be considered for publication after major revision in Sustainability. Prior publication of this manuscript following points needs to be addressed:

  • The rationale and strategy for the research design were not sufficient, and had critical flaws. The introduction should characterize the problem from the scientific side and not contain political motives. Introduction in general should be revised.
  • Figure 1 is a very interesting generalization, but it is of poor quality. It should be improved.
  • In Table 1, the "Data sources" and "Source of indicators" should be given as references. This approach is universally acceptable and improves the perception of the material.
  • It would be good to broaden the Discussion in the context of comparing the obtained results with the data of similar studies, especially, if possible, with other countries. The above discussion is of a local nature, and it does not correspond to the level of an international journal.
  • The topic discussed by the authors is quite debatable and can have many directions for development. Therefore, the approach proposed by the authors may have some limitations. To avoid this problem, I propose to add separate section "Limitations and prospects for further study".
  • References list should be carefully checked and journal style policy should be strictly followed (citation rule for books, monographs and articles, journal abbreviations, doi, etc).
  • Extensive English changes required.

My decision is major revision.

Author Response

Thank you for taking the time to read this article and expressing your recognition of the author's research.

Below, we will answer your comments and questions one by one:

Firstly, the issue in the introduction section. The authors have made comprehensive adjustments in content and structure. The introduction section elaborates on the practical background of the coupling and coordination between financial technology and digital rural areas from a scientific perspective, and proposes the importance and practical significance of the article based on theory and practice, as well as the innovation and contribution points of this article.

Secondly, the problem with charts. According to the reviewer's suggestion, remake Figure 1 using Visio software. And retained the data sources and indicator sources in Table 1

Thirdly, the issue of data and material comparison. Based on the comments of the reviewers and incorporating the characteristics of the text, we have chosen to include international data and experience comparisons in the introduction and results recommendations sections in order to enrich the diversity of the article.

Fourthly, limitations and future directions. Based on the comments of the reviewers, list the limitations of writing a separate section and the main directions for future research.

Fifth, regarding the formatting issues in the references, the authors have adjusted the content and format of the references.

Sixth, English modifications have been made.

Reviewer 5 Report

Comments for authors

The writing of the article was unclear and objective, and the structure of the article is also not the best (Introduction, literature review, methodology, results and conclusions).

In the introduction, it was not clear the main objective of the article or the secondary objectives. The authors need to define the research hypotheses and highlight the relevance of the study.

Better identify the originality and contribution of the study in the introduction, as well as, highlight the main results and their theoretical and practical implications.

In the literature review, the authors should bring previous studies related to the topic, as well as critique this literature.

The meaning of acronyms should be clear when they first appear, for example, what does CPC mean? In equation 11, what is the meaning of the variables included in the model?

The methodological aspects should present as much information as possible, which facilitates the replication of the study.

Before panel data is implemented, the authors should perform model fit tests, such as the Hausman test, to determine the best model fit for the data. On the other hand, the panel model results are only accepted after post-estimation tests for heterogeneity, correlation, and endogeneity.

The selection of variables should be presented after clarifying the sources, on the other hand, the selection of each variable and the expected impact should be reasoned and supported based on previous studies.

The results have not been clearly presented and need to be discussed, with the previous studies.

In line 586 the authors should rectify, where they mention that the value of the fixed effect is 0.960, to the value of the coefficient of determination of the fixed effects model, or, the value of the fit of the fixed effect model is 0.960.

In the conclusions, the authors should highlight the implications of the study, the limitations, and suggestions for future studies.

Author Response

Thank you for taking the time to read this article and expressing your recognition of the author's research.

Below, we will answer your comments and questions one by one:

Firstly, the issue of article structure. Revise the structure of the paper to include an introduction, literature review, methodology, results, and limitations based on the comments provided by the reviewers.

Secondly, the questions in the introduction. The authors have made comprehensive adjustments in content and structure. The introduction section elaborates on the practical background of the coupling and coordination between financial technology and digital rural areas from a scientific perspective, and proposes the importance and practical significance of the article based on theory and practice, as well as the innovation and contribution points of this article.

Thirdly, literature review and issues in theoretical mechanisms. This section has undergone structural and content adjustments. According to the reviewer's comments, the literature review was merged with the theoretical mechanism, and research hypotheses were proposed in the theoretical mechanism section, emphasizing the relevance of the research and the main and secondary objectives of the research. In the literature review, previous literature related to the subject was strengthened, and its marginal contribution to the field was evaluated and proposed.

Fourthly, the issue of abbreviations and variable names. According to the requirements of the reviewer, modifications have been made by adding the sources of variables, providing a detailed description of the possible effects of variables, and pointing out that previous research provides support for this article. At the methodological level, it provides as much information as possible to facilitate the replication of research.

Fifth, the issue of empirical content. This section adjusted the research process and results of the influencing factors. First, descriptive statistics and correlation analysis are added to show the basic situation of variable data. Hausman test is conducted before estimating panel data, and two-way fixed model is selected for estimation. Secondly, conducting regression analysis of the main model mainly focuses on the accuracy of hypotheses and conducting endogeneity tests to determine their stability. Finally, the samples were geographically divided to explore their heterogeneity.

Sixth, the issue of variables has already been discussed in the premise, so it will not be repeated here.

Seventh, the issues in the Results and Recommendations section. According to the reviewer's comments, the research significance of this article has been added in the conclusion section, and policy recommendations have been proposed to enhance the development of China's financial technology and digital rural areas.

Eighth, we have noticed the issue of fixed effect language, but in order to ensure the integrity of the article, the description in this section has been deleted.

Ninth, limitations and future research aspects. Based on the comments of the reviewers, list the limitations of writing a separate section and the main directions for future research.

Reviewer 6 Report

I think the paper is nice and well-organized. I just think that the authors could revise the language to make it easier to read, especially the contribution section. I want readers make it stronger. 

Author Response

Thank you for taking the time to read this article and expressing your recognition of the author's research.

Below, we will answer your comments and questions:

According to the comments of the reviewers, the authors have made revisions to the language of the article.

Round 2

Reviewer 4 Report

The authors mainly took into account the comments and significantly improved the manuscript. My decision is accept.

Author Response

Thank you again for the valuable feedback provided by the reviewers. I hope to provide you with valuable research papers next time!

Reviewer 5 Report

The authors complied with the suggestions left in the first review and this completely improved the article, however, in the discussion section they should with for their results based on the previous studies. How do they converge or diverge?

Author Response

Thank you very much for your feedback! The research team made further revisions by adding a convergence test in section 4.4 of the paper to examine the convergence and divergence of the coupling and coordinated development of fintech and digital rural areas, and drew corresponding conclusions.